# Statistical Analysis of Chemical Element Compositions in Food Science: Problems and Possibilities

**DOI:** 10.3390/molecules26195752

**Published:** 2021-09-23

**Authors:** Matthias Templ, Barbara Templ

**Affiliations:** 1Institute of Data Analysis and Processe Design, Zurich University of Applied Sciences, Rosenstrasse 3, CH-8401 Winterthur, Switzerland; 2Data-Analysis OG, AT-1110 Vienna, Austria; barbara.a.templ@gmail.com

**Keywords:** composition of food, log-ratio analysis, PCA, classification, artificial neural networks, adulteration, honey, saffron, chemical profiling

## Abstract

In recent years, many analyses have been carried out to investigate the chemical components of food data. However, studies rarely consider the compositional pitfalls of such analyses. This is problematic as it may lead to arbitrary results when non-compositional statistical analysis is applied to compositional datasets. In this study, compositional data analysis (CoDa), which is widely used in other research fields, is compared with classical statistical analysis to demonstrate how the results vary depending on the approach and to show the best possible statistical analysis. For example, honey and saffron are highly susceptible to adulteration and imitation, so the determination of their chemical elements requires the best possible statistical analysis. Our study demonstrated how principle component analysis (PCA) and classification results are influenced by the pre-processing steps conducted on the raw data, and the replacement strategies for missing values and non-detects. Furthermore, it demonstrated the differences in results when compositional and non-compositional methods were applied. Our results suggested that the outcome of the log-ratio analysis provided better separation between the pure and adulterated data and allowed for easier interpretability of the results and a higher accuracy of classification. Similarly, it showed that classification with artificial neural networks (ANNs) works poorly if the CoDa pre-processing steps are left out. From these results, we advise the application of CoDa methods for analyses of the chemical elements of food and for the characterization and authentication of food products.

## 1. Introduction

The importance of food composition data to nutrition and public health has been long acknowledged [1]. Currently, hundreds of articles have been published on the chemical composition of various kinds of food. The statistical techniques most often used are cluster analysis, principal component analysis (PCA), numerous classification methods, regression [2,3,4] and partial least-squares regression methods [5,6].

An inspection of the literature on the analytical and statistical methods frequently used in food science [2,3,4] as well as in chemometrics of honey [7] do not mention compositional data analysis (CoDa) [8]. A composition is the quantified decomposition of a whole into its component parts. Historically, a composition was described as random vectors with strictly positive components that added up to a whole, e.g., 100. Currently, it stands for all vectors that represent parts of a whole and carry relative information. The whole may only exist theoretically and be different for each composition [9]. CoDa, including the log-ratio methodology described later, is a method for describing the parts/connections of a whole that conveying relative information. Compositional methods are well established in many fields dealing with compositional data, such as material science [10], water chemistry [11], geochemistry [12], and air pollution chemistry [13]. Recently, the successful application of CoDa was demonstrated in food chemistry [14] by analyzing the chemical compounds in beer samples. It is well-known from the literature [9,15,16,17], that if traditional statistical analysis is applied to compositional datasets, correlations will be arbitrary [9,14] and even the arithmetic mean is not an adequate measure for the center of the distribution [18], which may lead to wrong conclusions [9,14,15]. Therefore, CoDa can be a way to gain additional insight and see beyond a constrained space (the simplex). While in most articles non-compositional methods for the statistical analysis of food are applied, there are a few exceptions. Cayuela-Sanchez (2020) used CoDa to investigate the composition of various pastries, biscuits [19] and olive oil [20]. Furthermore, E. Parent and their lab use CoDa theory for the diagnosis of various nutrients [21], for instance, fruit crops [22,23], bananas [24] and citrus [25]. Compositional data analysis focuses on log ratios between the parts (see Equations (Equation 3) and (Equation 4) for isometric and centered log-ratios), so that their relative scale and inherent interplay are accounted for. To demonstrate problems that may arise during the analysis of chemical elements in food science, datasets on the chemical compositions of honey and saffron were selected. Chemical profiling of honey [26,27] and saffron [28,29] is an important issue when determining their botanical and geographical origins. Honey is mainly composed of sugars and water with minor amounts of minerals, vitamins, amino acids, organic acids, flavonoids and other phenolic compounds, and aromatic substances [27,30]. The determinants of its composition, color, aroma and flavor are the flowers, geographical regions, climate and species of honeybee [30,31]. As mislabeling and adulteration of honey has become a worldwide problem, it is crucial not only to detect the adulterants in honey but also to classify honey samples correctly. The technical challenge of detecting adulterants in honey is widely discussed [7,32,33], the challenge of finding a theoretically sound statistical analysis is little understood. Similarly, saffron, which has numerous health benefits and is the world’s most expensive spice, is the object of fraudulent production and unethical trade practices [34]. The three major secondary metabolites which are important for the high quality of saffron are: crocins, which account for the yellow pigmentation from the stigma; picrocrocin, which gives it its rusty, bittersweet flavor; and safranal, which lends an earthy fragrance to the spice. It was hypothesized that additional insights into the chemical composition of honey and saffron samples might be obtained from a better interpretable results using CoDa. It was also assumed that a higher misclassification rate, lower predictive power, and a lower explained variance were inherent in a non-compositional analysis of compositional datasets.

The aim of this research was to compare compositional data analysis with classical statistical analyses to demonstrate how data pre-processing can influence a multivariate analysis, how a proper analysis can improve interpretation, and how a compositional method improves the accuracy of classification.

## 2. Results

Figure 1 shows the first two principal components through biplots of PCAs obtained from honey samples (see Section 4.1 for more information on the dataset used). It shows how the proportions of eigenvalues of the correlation matrix for the first two principal components differed depending on the type of data pre-processing. Figure 1A–C did not consider the special, compositional character or dependencies of such data using non-CoDa approaches. In more technical terms, statistical methods based on Euclidean geometry were applied to compositional data defined on the simplex. Figure 1D demonstrates the case when PCA was applied on centered log-ratio coordinates, which was in line with the principles of CoDa.

For the results in Figure 1A, the honey and syrup data were standardized. It was noticeable that all loadings pointed in a radius of less than 180°, i.e., more or less in the same direction. The reason is that the results were biased, since the correlations (as an essential input into the PCA) are strongly biased towards negativity [9,14,17,35], as was discovered by Pearson in 1897 [36]. The result is also difficult to interpret because not even the syrups were separated from the pure and adulterated honey observations. We also see such a bias in Figure 1B where the data were first logarithmized and then standardized (by [37]) to allow for equal influence of all variables. This result was also difficult to interpret, as, for example, it did not show any loadings to the syrup samples. We no longer see the previous kind of bias in Figure 1C, where the data were first brought to row sum 1 and then standardized to allow for equal influence of the variables. The variance of Figure 1C was much smaller (PC1: 24.9%, PC2: 20.7%) than found for Figure 1D (PC1: 36.1%, PC2: 20.0%). In Figure 1D the arrows indicating the elements of the honey samples are no longer distorted in the half-space, and the cosines of the angle between the arrows approximates the correlation between the log-ratio coordinates. It is clear that most variation in the data was explained by PC1; thus, not only aluminium (Al) and strontium (Sr) but also calcium (Ca) and manganese (Mn) had almost identically centered log ratios. Even most of the non-adulterated honeys pointed in the same direction on the biplot as the adulterated honeys (e.g., ACA and CA both showed highly negative scores for the first PC), but their magnitude was different. Furthermore, we see in Figure 1D that the adulterated honeys were separated from the non-adulterated ones. For example, all honeys of the type SS (*T. cochinchinesnsis*) are characterized by very high relative values of Mn, and the adulterated honeys are even higher. Similarly, other honeys, such as CA (*Chaste*), have relatively high values of Na. Again, this is more extreme with adulterated honeys. Syrup samples are characterized by relatively large values of Al and Sr.

The explained variance of the first two principal components is shown in Figure 2.

The explained variance of the first two principal components is highest when PCA was applied to pivot coordinates (46.3%, see “ilr” in Figure 2), followed by the application of the centered log-ratio coordinates (39.7%, see “clr” in Figure 2). This was also true when looking at other components, whereas this was only one of high interest for the first 1–4 components. Note that only about 46.3% of the variance in the dataset was explained by the first two components. It could be argued that this was not a high number and that caution should be used when interpreting the results because of the relatively high unexplained variance. Furthermore, it was also of interest to examine other components in biplots, e.g., to plot the first against the third component. Note also that the variance in the raw, unstandardized, untransformed dataset was almost 90% for the first two components (results not shown), but this was only related to the fact that the first two components were basically the two variables with the largest range of values, which was an uninteresting result. In other words, the explained variance is important, but many other aspects are also important (e.g., if the scores separate well). In any case, using a non-compositional treatment of the data loses a lot of explained variance, and the results might be arbitrary because the concept of an Euclidean metric in a simplex is not a proper concept.

For the saffron samples, their origins (Spain and Iran) among other things were compared (Figure 3), but the same problems as we saw in Figure 1 arose but even more clearly. Standardization without transformation gave very poor results, and the negative bias was omnipresent (see Figure 3A). This could also be seen in the log-transformed and standardized data in Figure 3B. In addition, the separability left a lot to be desired in Figure 3C. On the other hand, the biplot obtained by PCA of the centered log-ratio coordinates (Figure 3D) showed a clear separation between the Spanish and Iranian saffron samples. The Spanish saffron had higher relative concentrations of lead (Pb) and cobalt (Co) and was therefore more likely to be contaminated with these two elements.

The explained variance is highest with an isometric log-ratio transformation (see Figure 4), but with this transformation, the interpretation of the results was more difficult. However, the centered log-ratio results were usually better than the non-compositional methods.

The results of the PCAs clearly showed the benefits of using CoDa, while the classification results were more diverse. Figure 5 shows the average misclassification rate quantified for different classification methods. The target variable for the honey samples holds the information about whether the observation was of raw honey, a syrup or an adulterated honey. Obviously, compositional data analysis outperformed most of the non-compositional approaches, but the results on log-transformation and log-transformation plus standardization were comparable. The reason is that the larger the number of variables, the less the log transformation differs from a (centered) log-ratio transformation. The denominator for high-dimensional compositional data usually shows hardly any data structure, only noise. This means that the main role in the observations was played by the dominant components in the log ratios. From a more technical point of view, the log ratio of the geometric means in the comparison of the log and Aitchison distances was almost 0 despite the large number of variables. We refer to arceló-Vidal et al. [38] page 189, Equation 14.1. Even if from a theoretical point-of-view the simple log-transformation is wrong [39], the results might be comparable when the dataset consists of many parts. Interestingly, the variant of the pivot log-ratio transformation (*ilr_var*) gave slightly better results than without any ordering of parts. As the honey dataset also contained non-detects, various replacement strategies were also considered. The replacement method based on compositional methods (method bdls_pls [40]) clearly gave the best results in subsequent classifications.

Figure 6 shows the results for the saffron samples. Since the dataset was complete, there was no need to apply a replacement methodology, as was the case for the honey samples (see Figure 5). When the pivot coordinates (ilr) were classified, more or less, the least misclassification instances were received; however, the centered log-ratio and simple logarithmic transformation gave a similar amount of misclassification instances. Artificial neural networks are to be treated with caution here since we are dealing with a very small dataset and these methods only work well with somewhat larger data. A slight overfit took place using ANNs that could not be avoided by reducing the complexity of the network or by introducing a higher dropout rate.

A closer look at Figure 6 reveals that the results for the log transformation were similar to the CoDa results. In fact, the log transformation (although not the right choice for compositions) was similar to the log-ratio transformation when the number of parts was high (see also the argumentation before for the honey dataset). This was exactly the case with the saffron data. One should note that CoDa has other advantages besides the misclassification rate, e.g., normalization is no longer necessary.

## 3. Discussion

Compositional data analysis using log-ratios is a theoretically sound concept that is well known in many sciences but rarely applied in food science. It is problematic because, if traditional statistical analysis is applied to a compositional dataset, correlations can be arbitrary and even the arithmetic mean is not an adequate measure for the center of the distribution [9]. Both of our null hypotheses for interpretability and misclassification rates were supported: higher explained variance and smaller misclassification rates were obtained when the compositional nature of the datasets was considered in the analysis.

Whenever a method takes the nature of compositional data into account, it leads to better interpretability of the results. Biplots obtained from various PCAs demonstrated how the pre-processing of data may influence the analysis. When CoDa was not considered, biplots were clearly distorted, which was best seen from the direction of the loading vectors. This is because the concept of linear correlation was not working and was theoretical unsound since the correlation between the parts of a composition is always biased toward a negative one. The variance of the first principal components was the highest when clr and ilr were applied, which confirmed that it was not advisable to apply PCAs to compositional data without using an appropriate log-ratio presentation of the data. The highest misclassification instances were gathered when no transformation was performed before data analysis or when the data were closed and standarized. Thus, the accuracy of the classification methods improved when CoDa was used.

To sum up some advantages, the theoretical correctness of compositional data analysis methods is undoubted and has been proven by many authors starting with the main works of [8]. In addition, the size effect—when a true measurement (e.g., an instrumental signal) $x=\left[x_{1},x_{2},\ldots ,x_{n}\right]$ cannot be observed directly but $\mathrm{cx}=\left[{\mathrm{cx}}_{1},{\mathrm{cx}}_{2},\ldots ,{\mathrm{cx}}_{n}\right]$ is observed—can be ignored when using compositional data analysis. The measurement is from the same equivalence class and the ratios between parts are also the same. Higher predictive power and better results are generally obtained.

Note that class modelling approaches can also be used for classification with respect to a one-class classification problem [41], such as when investigating adulterated versus non-adulterated honey or genuine honey versus all other non-real honey samples. One way to do this is classical soft independent modelling by class analogy (SIMCA) [42] or robust SIMCA [43]. The results were not satisfactory and the three other methods (LDA, KNN, and ANN) outperformed SIMCA, so the results were excluded so as not to go beyond the scope of the paper.

However, there are also several drawbacks to be discussed. Outliers are produced after presenting data in centered or isometric log-ratio coordinates whenever an observation lays on the boarder of the simplex. One solution is to use robust statistical methods to analyse such data [9]. True zeros and rounded zeros are not in the simplex by definition and a log-ratio with a zero is not possible. True zeros are still an unsolved problem in CoDa even though some solutions have already been presented [44]. Rounded zeros often come from too-small concentrations with too few precise measurement units. Rounded zeros cause extra work when using compositional methods, and they must be imputed first by using a censored method. Solution by imputation of rounded zeros are outlined in this contribution [40,45] and were applied to the honey samples. In addition, the centered log-ratio transformation is often used because of its simplicity, but using well-selected balances [8] for specific isometric log-ratio transformations often leads to better interpretable results. In addition, for instrumental signals (e.g., NMR, LC-MS, or GC-MS), all possible log-ratios may be used instead of centered or isometric log-ratios [46]; that is, each variable is divided by one of the other variables before the logarithm is taken. For a dataset with 10 variables, there are already 45 possible log-ratios between the variables. The authors of [47] suggested using all possible log-ratios, but since this would lead to a large number, they suggest using feature selection to reduce the number of log-ratios. They argue that a centered log-ratio transformation may average too much leading to a higher false discovery rates of biomarkers [47].

This study had limitations. The usage of CoDa was demonstrated only on two datasets (honey and saffron), which originated from different fields of food science (food substance and spice). Therefore, other food datasets need to be analyzed with CoDa to establish its broad usage in food science. Furthermore, for the application of ANNs we would have needed larger datasets as these methods work better with big data, but in the field of food science, large datasets are seldom available. However, our results indicated that by incorporating the theory of CoDa, the predictive classification methods will lead to better performance, which may be used to improve the characterization of food products.

Our aim was to create awareness of the choice of compositional methods when compositional data to be analyzed. The CoDa of mineral elements of honey samples as well as trace element concentrations of saffron samples allowed us to demonstrate the correct assessment of compositions and to recommend that this application be extended to an analysis of any food composition. It would also allow for the establishment of CoDa in food science. It is expected that similar results can be gathered from the analysis of other datasets of other food substance and spices.

## 4. Materials and Methods

### 4.1. Mineral Element Data of Honey Samples

A total of 201 pure honey and 45 syrup samples from local beekeepers, specialized markets, factories and supermarkets of various botanical and geographical origins in China and Mongolia were collected and analyzed by Liu et al. [37]. Luo [48] published the mineral profile of 6 types of monofloral honeys, including (i) Acacia honey (*Robinia pseudoacacia* L., AC1–AC14), (ii) Chaste honey (*Vitex negundo* var. *heterophylla* (Franch.) Rehd., CA1–CA10), (iii) Jujube honey (*Ziziphus jujuba* Mill.var.inermis (Bunge.) Rehd., JU1–JU10), (iv) Linden honey (*Tilia amurensis* Rupr., LD1-LD14), (v) *Triadica cochinchinensis* honey (SS1–SS12), (vi) Rape honey (*Brassica napus*, RP1–RP7). The authors [37] added various syrup solutions to mimic adulterated honey samples. Thus, 183 adulterated honey samples were obtained by a standard method (Product No. A01-00047; see [37] for more details). Furthermore, 18 blind samples were tested by Agilent 5100 Synchronous Vertical Dual View ICP-OES. The chemical concentrations of the following 12 elements (mg/kg): Al, B, Ba, Ca, Fe, K, Mg, Mn, Na, P, Sr, and Zn were selected by Liu et al. [37] to be studied—7.75% of the values were non-detects or missing.

Liu et al. [37] analyzed the samples to distinguish between honey and syrup-based adulteration using principal component analysis and applied the sparse partial least squares discriminant analysis method to optimize the differentiated models of honey and adulteration by mineral element chemometrics profiling. In this study, we reanalyzed the data published by Liu et al. [37] using CoDa.

### 4.2. Stable Isotope Ratio and Trace Element Concentration Data of Saffron Samples

From Iran, 41 saffron samples were collected by the authors of Wakefield et al. [29] directly from producers in northeastern (Khorasan Province) between September 2010 and November 2011. Nine samples were collected from the La Mancha region in Spain in November 2011 by the authors directly from producers and a further 2 from a trusted commercial redistributor [29]. In the case of both datasets, concentrations of 42 elements were determined as described by [29]. Of the 42 elements, 29, namely Li, B, Na, Mg, Al, K, Ca, V, Mn, Fe, Co, Ni, Cu, Zn, Ga, As, Rb, Sr, Y, Mo, Cd, Cs, Ba, Ce, Pr, Nd, Sm, Gd, Pb, were analyzed by Wakefield et al. [29]. Welch two-sample *t*-tests as well as a principal component analysis and linear discriminant analysis (LDA) were applied to various data subsets by Wakefield et al. [29] to find an approach for the origin verification of saffron. In this study, the data from [29] was reanalyzed and made available on Mendeley Data [49].

## 5. Data Analysis

### 5.1. Non-Compositional Standardization of Variables

Whenever a non-compositional multivariate method is applied to data where the variables have very different ranges of values, it is well-known that the data should be standardized in advance [50] to ensure that each variable has approximately the same influence in the multivariate analysis. For example, if we apply PCA to the raw data, the result will be dominated by the variable K in the honey dataset because the values are many times (up to 2000 times) higher than the others.

For the standardization, so-called *z*-scores were used, i.e., variables were rescaled in such way that for each variable the arithmetic mean was 0 and the variance equaled 1. More precisely, from data matrix $X=\left\{\left(x_{ij}\right)\right\}\;(\mathrm{with}\;i=1,\ldots ,n;\;j=1,\ldots ,D$), we obtained the elements of the *z*-scores matrix Z by
$$z_{ij}=\frac{x_{ij}-{\bar{x}}_{j}}{s_{i}}\;,$$
with the arithmetic mean of the *j*th variable ${\bar{x}}_{j}=\frac{1}{n}\sum_{i=1}^{n}x_{ij}$ and the standard deviation of the *j*th variable $s_{j}=\sqrt{\frac{1}{n-1}\sum_{i=1}^{n}{\left(x_{ij}-{\bar{x}}_{j}\right)}^{2}}$.

### 5.2. Non-Compositional Standardization of Observations

Rescaling of compositions to a constant sum (e.g., 1 or 100) can be performed with a closure operator C. Consider a composition $x={\left(x_{1},\ldots ,x_{D}\right)}^{\prime }\in R_{+}^{D}$, where $R_{+}^{D}$ denotes the *D*-dimensional real space with strictly positive elements, so $x_{i}>0$ for i=1,…,D. The closure of x to any positive number κ is defined as
(1)$$C_{\kappa }\left(x\right)={\left(\frac{\kappa \cdot x_{1}}{\sum_{i=1}^{D}x_{i}},\ldots ,\frac{\kappa \cdot x_{D}}{\sum_{i=1}^{D}x_{i}}\right)}^{\prime }.$$

The parts of this new vector add up to the desired constant κ used to rescale the parts of a composition. By setting κ=1, a composition x with any arbitrary sum of parts is rescaled to a composition $C_{1}\left(x\right)$ with the component sum equal to one. However, this new vector is compositionally equivalent to the original vector; thus, compositional analysis provides exactly the same results [9].

Therefore, we applied such a rescaling/closure in our comparison. We refer to this methodology as closed. Since dominant variables with large values (e.g., the variable *K* in the honey sample) would still have a very dominant influence on the results, we also have to standardize the variables of the closed observations.

### 5.3. Non-Compositional Transformation

Transformation of the variables often helps to establish a linear relationship between the variables. We applied log-transformation in various situations as it is often used [51,52] before a non-compositional multivariate statistical method was applied.

### 5.4. Compositional Analysis

Compositional data are typically represented in proportions or percentages, but other units, such as chemical elements in parts per million (ppm), mg/kg, and mg/L, are used to reflect their relative nature. In our examples, the units of our measurements are in mg/kg. Relative information is the most important for a food composition; absolute numbers and the unit of measurement are less so and sometimes non-informative. If chemical compounds decrease or increase during a chemical process, such as the adulteration of honey over time, the decrease is measured on a relative scale, and it is proportional to the whole. Aitchison [8] recognized the importance of relative information and founded the principles of compositional data analysis.

The properties of compositional data can be summarized into three principles [9,14,53]: (1) scale invariance, which means that changing the scale of units does not affect the results such as limiting the observations to 1; (2) permutation invariance, which means that changing the order of variables does not affect the results; and (3) subcompositional coherence, which means that, (a) information conveyed by a composition of *D* variables should not be in contradiction with one coming from a subcomposition of fewer than *D* variables, and (b) adding more components does not influence the conclusion about any subcomposition. When non-compositional methods are applied to compositional data, these properties are not fulfilled.

The sample space of the food compositional data is defined (after [9]) as
(2)$$S^{D}=\left\{x={\left(x_{1},\ldots ,x_{D}\right)}^{\prime }\in R_{+}^{D}|\forall \kappa >0\exists !\lambda >0:x=\lambda C_{\kappa }\left(x\right)\right\}$$

### 5.5. Standardization and Transformation by Means of Log-Ratios

The most coherent way of analyzing compositional data is by applying a log-ratio analysis, i.e., applying classical statistical methods on log-ratio coordinates. The aim of log-ratio analysis is to find an orthogonal representation (log-ratio coordinates) of the compositional data in Euclidean space. A composition can be mapped from $S^{D}$ to the real space $R^{D-1}$ using an (isometric) log-ratio transformation. One possible representation of compositional data in $R^{D-1}$ is pivot coordinates [9], which are based on the isometric log-ratio coordinate representations [53] and form a special orthogonal basis, given by
(3)$$ilr\left(x\right)=z={\left(z_{1},\ldots ,z_{D-1}\right)}^{\prime }\;\mathrm{with}\;z_{j}=\sqrt{\frac{D-j}{D-j+1}}\log\frac{x_{j}}{(\prod_{k=j+1}^{D}x_{k}{)}^{1/\left(D-j\right)}}\;,$$
for j=i,…,D−1. The first variable $x_{1}$ only appears in coordinate $z_{1}$, while $x_{2}$, for example, appears in both $z_{1}$ and $z_{2}$. A variable $z_{j}$ can thus be interpreted as its relative dominance with respect to the geometric mean see Equation (Equation 6) of the other remaing j+1 variables. We denote this method as *ilr* in the following.

A minor variant of Equation (Equation 3) is denoted by *ilr_var*. Here, the parts are ordered according to their correlation to the logarithm of the parts/exlanatory variables with the target variable. The ordering of parts is done from the highest to lowest correlation.

Centered log-ratio coordinates represent a popular orthogonal representation of compositions in Euclidean space, chosen mostly because of its simplicity. They can be obtained by applying a centered log-ratio transformation, namely, by dividing each value of a composition of the geometric mean of the whole composition and then taking the logarithm. Thus, a composition $x\in {\tilde{S}}^{D}$ can be expressed by the vector $y\in R^{D}$, with
(4)$$\mathrm{clr}\left(x\right)=y={\left(y_{1},\ldots ,y_{D}\right)}^{\prime }={\left(\ln\;\frac{x_{1}}{\sqrt[{D}]{\prod_{k=1}^{D}x_{k}}},\cdots ,\ln\;\frac{x_{D}}{\sqrt[{D}]{\prod_{k=1}^{D}x_{k}}}\right)}^{\prime }.$$

For an n×D matrix X of compositional data with the compositions $x_{i}^{\prime }=\left(x_{i1},\ldots ,x_{iD}\right)$ in the rows of X, the *n* rows of the matrix of clr coefficients Y is obtained by
(5)$$y_{i}^{\prime }={\left(\mathrm{clr}(x_{i})\right)}^{\prime }=\left(\ln\;\frac{x_{i1}}{\sqrt[{D}]{\prod_{k=1}^{D}x_{ik}}},\cdots ,\ln\;\frac{x_{iD}}{\sqrt[{D}]{\prod_{k=1}^{D}x_{ik}}}\right).$$

The denominator used in Equation (Equation 4) is the geometric mean, and the product can be also represented in logarithmic representation for reasons of higher numerical accuracy,
(6)$$g_{m}\left(x\right)=\sqrt[{D}]{\prod_{k=1}^{D}x_{k}}=\exp\left(\frac{1}{D}\sum_{k=1}^{D}\ln\;x_{k}\right).$$

Note that the geometric mean used in the denominator of Equation (Equation 5) is calculated for each observation.

It is easy to see that zeros caused problems in Equations (Equation 3)–(Equation 6). Zeros are not included in the simplex see Equation (Equation 2) and must be replaced in advance, see Section 5.6 for more details.

Centered log-ratio coordinates are often used in compositional biplots because of their simplicity and symmetry [54,55]. For the PCA, we evaulated the data samples using centered log-ratio analysis [8], and for classification we also compared them to pivot coordinates. When interpreting the results of a CoDa, $S^{D}$ was estimated as the space of the composition and $R^{D-1}$ as the Euclidean space where the methods were applied to log-ratio coordinates.

### 5.6. Replacement of Missing Values and Non-Detects

Chemical compositions sometimes contains missing values (see [56] for methods to deal with it) and often contain rounded zeros from non-detects. Zeros in compositional data are classified into “essential”, or true, zeros and “rounded zeros” [40], and strategies to deal with them are needed in a CoDa [40,57]. Rounded zeros, values below the detection limit, occur more frequently in chemical compositions of food. Several advanced “rounded zero” replacement strategies have been suggested to deal with this problem [40,58,59,60] because of the special nature of compositional data.

We compared various methods using the honey samples (see Section 2) to show the goodness of different (non-compositional and compositional) replacement strategies for non-detects. Specifically, we calculated the misclassification rates of the classification methods using different replacement strategies. Only one replacement method was used for the PCA (method *dl23*, see below for details).

The following non-compositional strategies were used to replace rounded zeros:**const:** Any rounded zero value is replaced by a **constant** value of 0.1. Note that it is not a good strategy to impute rounded zeros. However, this method should serve as a benchmark, among other things.**dl23:** This comparatively equally simple method also replaces all zeros with a constant value smaller than the **two-thirds of the detection limit**. Martín-Fernández et al. [58] found that the detection limit minimizes the distortion in the covariance structure.**unif:** A zero is replaced in a variable $x_{j}$ by drawing a random **uniform** number between the interval $\left[0.1\cdot min\left(x_{j}^{\left(+\right)}\right);0.9\cdot min\left(x_{j}^{\left(+\right)}\right)\right]$, with $x_{j}^{\left(+\right)}$, the smallest positive value of variable *j*. It prevents a zero being imputed to close to 0 and ensures imputation below an unknown detection limit.

In addition, the following compositional strategy was used to replace rounded zeros:**bdls_pls:** (**b**elow-**d**etection-**l**imit using (censored) **p**artial **l**east **s**quares regression) A zero is replaced by an iterative EM-algorithm based on a censored partial least squares estimation on sequential log-ratio coordinate representations. For details, see [40].

### 5.7. Principal Component Analysis

Principal component analysis [61] allows one to obtain new orthogonal projections of the original data based on the maximum variance of the projected data. Namely, the first principal component (the first score vector) is the linear combination of the variables of the dataset for the projected values having the largest variance, which explains most of the data. The second principal component is the linear combination of the variables with two restrictions: being the second-highest variance explained and being orthogonal to the first principal component. A PCA is frequently applied to characterize the components of food chemical compositions see, e.g., [29,37] and typically serves as an exploratory method to interpret the multivariate dependencies in the dataset. For the latter case, the resulting first two (orthogonal) PCs are often visualized in a biplot that allows visualizing the magnitude and sign of each variable’s contribution to the first two principal components.

Biplots from PCAs were applied on (1) standardized compositional data, (2) log-transformed and standardized compositional data, (3) closed and standardized data, and (4) centered log-ratio coordinates to show the variations in chemical components of the honey and saffron samples.

### 5.8. Classification

To confirm the performance of compositional analytical over classical methods, the average misclassification (misclassified observations expressed in percentages) was quantified for different classification methods applied on the dataset modified in the following ways:zeros replaced with const, dl23, unif, and bdls_pls (see Section 5.6).no transformation, standardization, log-transformation, log-transformation and standardization, rescaling by closure, or pivot coordinate or centered coordinate representation.

Three types of classification methods were applied: LDA, KNN, and ANN. Linear discriminant analysis (LDA) [62] is a supervised classification method based on normality assumptions for separating the groups. *k*-nearest neighbor classification (KNN; [63]) classifies a data point based on the class of its *k* nearest neighbors. Finally, a deep artificial neural network (ANN) [64] was chosen as representative of a non-linear classification method. Neural networks represent non-linear statistical models based on weighted linear combinations of observed values and their activation by a non-linear transformation. Millions of weights are adjusted to obtain the best possible output (according to a loss function and evaluation metric) from input data and multiple layers. The weights (neurons in a network) are iteratively improved by a stochastic gradient method.

The ANN is used with the following parameter settings (see also [45]):20% validation/80% training data,3 layers, 300 neurons in the first layer, followed by 128 and 64 neurons in the next layer,10% dropout in the first 2 layers,adam optimizer [65] and activation function reLu [66],mean squared error as a loss function and mean absolute error as an evaluation metric, and500 epochs with break whenever 50 epochs do not improve the result

Note that (many) other parameter settings have also been tested (up to 10 layers and more than 3 million trainable parameters). To estimate the misclassification rate of each method, a 10-fold cross validation (repeated 5 times and averaged) were used to train the models on the training sets and to evaluate them on the test sets. We did not use an additional truly external validation dataset.

All analyses were performed using the software and environment R [67]. Data were visualized via the R package ggplot2 [68]. The R packages robCompositions [69] were used for log-ratio transformations, replacement with bdls_pls [40] and principal component analysis [9]. The R package caret [70] was used for KNN classification and package MASS [71] for linear discriminant analysis. As an interface to keras and tensorflow, the keras R-package [72] was used for artificial neural networks.

## 6. Conclusions

Principal component analysis revealed the pitfalls of classical analysis conducted on compositional data: distorted biplots and less-explained variance. Classification resulted in a less predictive power when a non-CoDa method was used. Replacement strategies of non-detects should be also based on log-ratio methods. Generally, using CoDa for chemical elements not only resulted in higher explained variance and lower misclassification rates but also enabled better interpretability of the results. However, depending on the type of data, one can expect some difficulties, which are mentioned in the discussion section (outliers, the zero problem, and the choice of log-ratio transformation). It is therefore advisable to apply compositional analysis (CoDa) methods in the analysis of chemical elements in food.

## Figures and Tables

**Figure 1 molecules-26-05752-f001:**
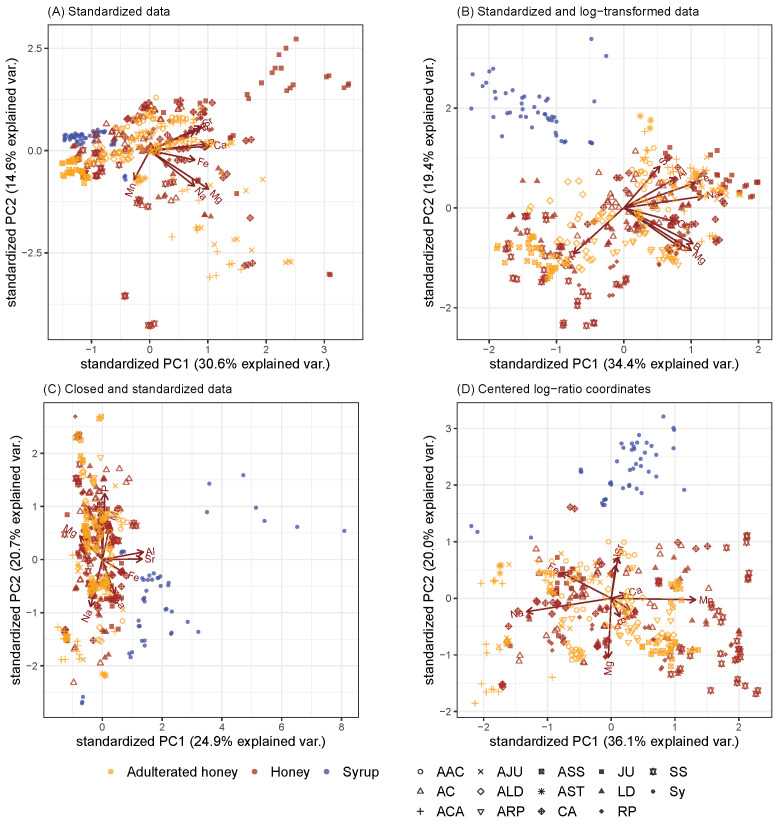
Biplots obtained from honey samples (pure and adulterated). First, two principal components represented by biplots of the PCA applied on (**A**) standardized data, (**B**) standardized and log-transformated data, (**C**) closed and standardized data, and (**D**) centred log-ratio coordinates. Abbreviations of various type of honey: AC: *Acacia*, CA: *Chaste*, JU: *Jujube*, LD: *Linden*, SS: *T. cochinchinensis*, RP: *Rape*; and various types of sugar syrups: Sy; and adulterated honey categories: AAC (adulterated *Acacia*), ACA (adulterated *Chaste*), AJU (adulterated *Jujube*), ALD (adulterated *Linden*), ARP (adulterated *Rape*), ASS (adulterated *T. cochinchinensis*).

**Figure 2 molecules-26-05752-f002:**
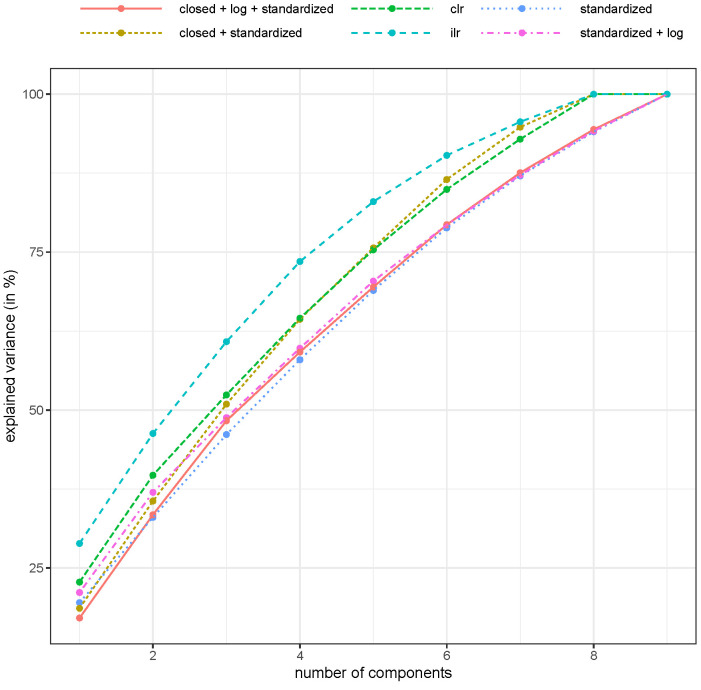
Explained variance (in %, cumulative) for different numbers of components and different pre-processing of the compositional honey samples. Abbreviations: clr: centered log-ratio coordinates, ilr: isometric log-ratio transformed data (i.e., pivot coordinates).

**Figure 3 molecules-26-05752-f003:**
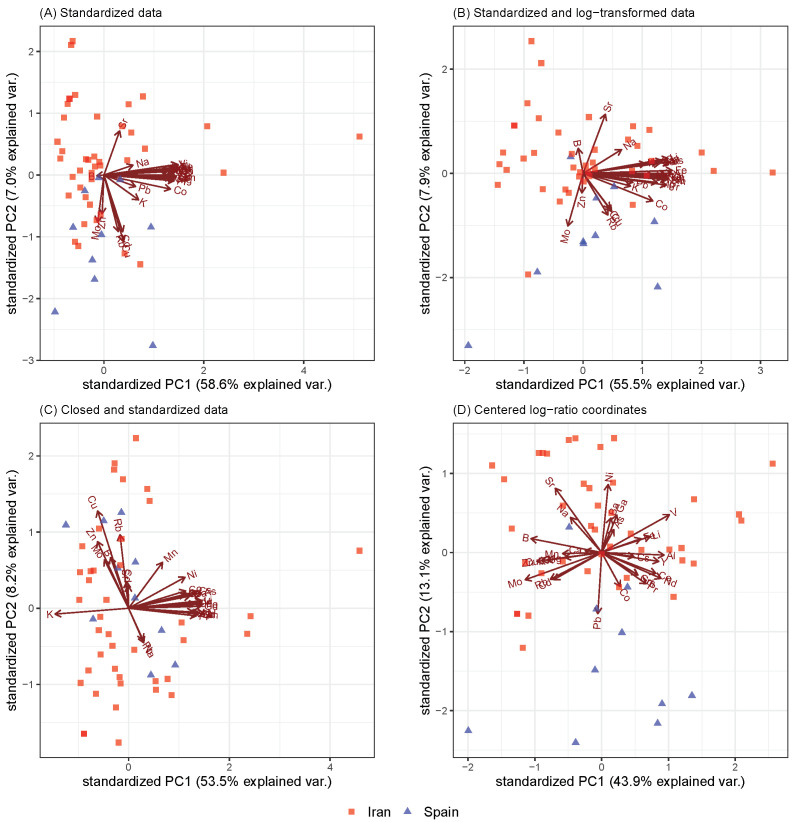
Biplots obtained from saffron samples originating from Iran and Spain. First two principal components represented by biplots of the PCA that was applied on (**A**) standardized data, (**B**) standardized and log-transformated data, (**C**) closed and standardized data, and (**D**) centred log-ratio coordinates.

**Figure 4 molecules-26-05752-f004:**
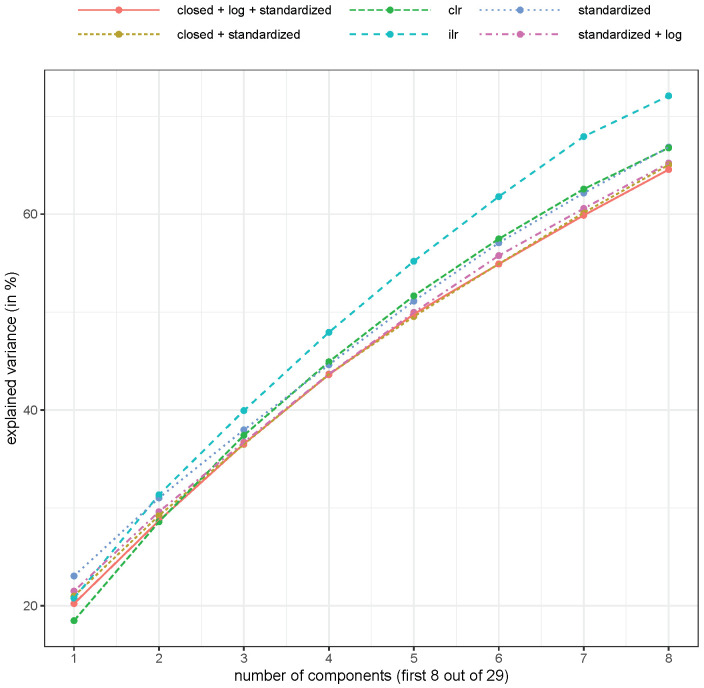
Explained variance (in %, cumulative) for different numbers of components and different pre-processing of the compositional saffron samples. Abbreviations as for Figure 2.

**Figure 5 molecules-26-05752-f005:**
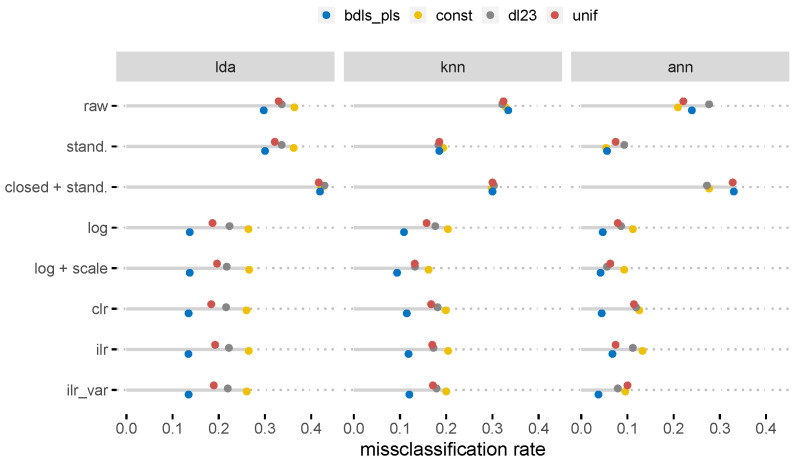
Misclassification rates of various classification methods based on different pre-processing and replacement strategies applied to the honey samples. Abbreviations (for details, see Section 3): lda: linear discriminant analyis, KNN: k-nearest neighbor, ANN: artificial neural network; bdls: below detection limit using (censored) partial least squares regression, const: constant, dl23: two-thirds of the detection limit, unif: uniform; closed + stand: closed and standardized data, raw: raw, i.e., non-transformed, log: log transformed, scale: scaled, ilr: isometric log-ratio transformed (i.e., pivot coordinates), clr: centered log-ratio coordinates.

**Figure 6 molecules-26-05752-f006:**
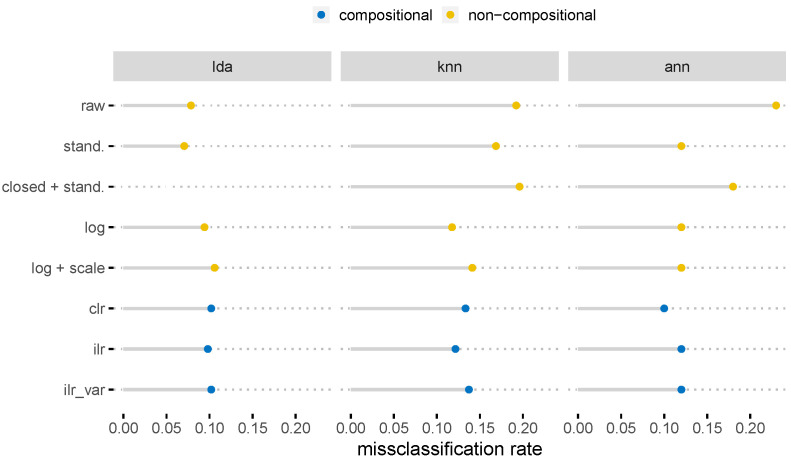
Misclassification rates based on different pre-processing of the saffron samples.

## Data Availability

Dataset License: CC0 1.0 (honey samples) and CC BY 4.0 (saffron samples). Both datasets are available for free (honey sample under licence CC0 1.0, saffron sample under licence CC BY 4.0) at [48,49], and they are also accessible via the R package robComposiitons [9].

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
