# Peer review of "Statistical Analysis of Chemical Element Compositions in Food Science: Problems and Possibilities"

_molecules, 2021, doi:10.3390/molecules26195752_

Round 1

Reviewer 1 Report

Authors satisfactorily answered to the reviewers' comments of the previous submission stages. I was already quite satisfied of the previous version of the manuscript and I think it has further improved.

Author Response

Thank you for your time, effort and positive feedback. We have improved the manuscript with regard to the English language.

Reviewer 2 Report

Thank you for the opportunity to review the paper. I hope that my comments are helpful to the authors in order to improve the quality of their manuscript for acceptance. Below find my comments, I try to be brief as much as possible.

Introduction

Line 32: I think that a more detailed definition of the CoDa methodology is required

Line 48: I would suggest to explaining the meaning of “log-ratio”

Line 60 and 61: “challenge” instead of “challenge”

Line 61: I would report only the references “[7,32,33]”, instead of “see [7,32,33]”

Line 63: comma is missing “…spice in the world, is regularly exposed…”

Line 67: only “safranal” instead of “and safranal”

Results

Line 78: I would add “…shows the first two principal components through biplots of PCAs obtained from honey and syrup samples”

Line 81: “…the special, compositional nature of such data…” please clarify

Line 128: I would add in this first sentence the origin of the Saffron samples compared (Spanish and Iranian)

Line 147: “The reason is that the larger the number of variables, the less the log transformation differs from a (centered) log-ratio transformation, because of the course of dimensionality” this sentence is not very clear. Please rephrase, if it is possible

Line 155: a bracket is missing (method bdls_pls [41])

Figure 1 and 3: the name of the elements in the plots are hardly visible; please zoom it, if it is possible, or use different colours

Discussion

Line 205: I would report only the references “[42]”, instead of “see e.g. [42]”

Line 212: “Especially, for instrumental signals (e.g., NMR, LC-MS, or GC-MS) [44] and especially [45] proposed to use all possible log-ratios before (possible) feature selection of the possible large number of log-ratios, because a centered log-ratio transformation may average too much leading to a higher false discovery rates of biomarkers [45]” this sentence is not very clear as well as the positioning of the references. Please rephrase, if it is possible

Materials and methods

Section 4.1: I would add more details about the analytical method applied

Line 237: “…analysed by [37]” I would report also the name of the authors, e.g. “…analysed by Liu et al. [37]”

Line 248: “…were selected by [37]” I would report also the name of the authors

Line 249: “…the authors of [37]” I would report also the name of the authors

Line 253: “…data published by [37]” I would report also the name of the authors

Line 255: “…by the authors of [29]” I would report also the name of the authors

Line 262: “…were analysed by [29]” I would report also the name of the authors

Line 264: “…by [29]” I would report also the name of the authors

Line 265: “…data from [29]” I would report also the name of the authors

Line 265: “…was reanalysed that was available as [47]” please clarify

Line 297: “…mg/kg, and “cancel the comma

Line 303 “[8] recognized” I would report also the name of the authors

Line 308: “…the results, and “cancel the comma

From Line 305 to 312: too long sentence, please consider to rephrase

Line 346 “…[41], and “cancel the comma

Line 347 “…i.e., values below…“cancel the comma

Line 350 “… [41, 55-57], considering “a comma missing after bracket

Line 361: “… [55] found” I would report also the name of the authors

Line 387: “…and (4) on centered log-ratio” instead of “and on (4) centered log-ratio”

Author Response

Thank you for the opportunity to review the paper. I hope that my comments are helpful to the authors in order to improve the quality of their manuscript for acceptance. Below find my comments, I try to be brief as much as possible.

Answer: Your comments were indeed very helpful. Thank you very much.

Introduction

Line 32: I think that a more detailed definition of the CoDa methodology is required

Answer: Thank you for this suggestion. We now introduced a composition more careful incuding more details, before describing the CoDa methodology naming the log-ratio analysis as one prominent kind of CoDa methodology. We changed the text accordingly.

Line 48: I would suggest to explaining the meaning of “log-ratio”

Answer: Thank you for this comment. The answer depends and is not as simple. We now refered to the Material and Methods Section, Equation 3 and 4. Here already both used log-ratio's are explained. We further discuss the pro and cons in the discussion.

Line 60 and 61: “challenge” instead of “challenge”

Answer: Thanks, done.

Line 61: I would report only the references “[7,32,33]”, instead of “see [7,32,33]”

Answer: Thanks, we considered and changed it for three cases.

Line 63: comma is missing “…spice in the world, is regularly exposed...”

Answer: Thanks, done.

Line 67: only “safranal” instead of “and safranal”

Answer: Thanks, done.

Results

Line 78: I would add “... shows the first two principal components through biplots of PCAs obtained from honey and syrup samples”

Answer: Thanks, done.

Line 81: “... the special, compositional nature of such data…” please clarify

Answer: we added: ... compositional character and compositional dependencies of such data using non-CoDa approaches. In more technical terms, statistical methods based on Euclidean geometry were applied to compositional data defined on the simplex.

Line 128: I would add in this first sentence the origin of the Saffron samples compared (Spanish and Iranian)

Answer: Thanks for this suggestion. It is now mentioned.

Line 147: “The reason is that the larger the number of variables, the less the log transformation differs from a (centered) log-ratio transformation, because of the course of dimensionality” this sentence is not very clear. Please rephrase, if it is possible

Answer: Is now rephrased and added: The denominator for high-dimensional compositional data usually hardly shows any data structure, but rather only noise. This means that the main role in the observations is then played by the dominance of the components in the log ratios, and, from a more technical point of view, the log-ratio of the geometric means in the comparison of the log and Aitchison distances is almost 0, despite a large number of variables. We refer to arcel\'o-Vidal et al. [38, page 189, Equation 1]. 

We did not include the proof in the paper, but here it is, whereby $\mathrm{d}_{e}^{2}$ is the Euclidean distance and $\mathrm{d}_{a}^{2}$ the Aitchison distance. Since the clr (and also isometric log-ratio transformations works with geometric means, this means that the log and centred log-ratio (or isometric log-ratio) solutions gets more similar the higher the number of parts/variables in a data set.)

$$
\begin{aligned}
\mathrm{d}_{e}^{2}(\log (\mathbf{x}), \log (\mathbf{y})) &=\sum_{i}\left(\log \left(x_{i}\right)-\log \left(y_{i}\right)\right)^{2} \\
&=\sum_{i}\left(\log \frac{x_{i}}{\mathrm{~g}_{\mathrm{m}}(\mathbf{x})}-\log \frac{y_{i}}{\mathrm{~g}_{\mathrm{m}}(\mathbf{y})}+\log \frac{\mathrm{g}_{\mathrm{m}}(\mathbf{x})}{\mathrm{g}_{\mathrm{m}}(\mathbf{y})}\right)^{2} \\
&=\sum_{i}\left(\log \frac{z_{i}}{\mathrm{~g}_{\mathrm{m}}(\mathbf{z})}\right)^{2}+2 \log \frac{\mathrm{g}_{\mathrm{m}}(\mathbf{x})}{\mathrm{g}_{\mathrm{m}}(\mathbf{y})} \sum_{i}\left(\log \frac{z_{i}}{\mathrm{~g}_{\mathrm{m}}(\mathbf{z})}\right)+D \log ^{2}\left(\frac{\mathrm{g}_{\mathrm{m}}(\mathbf{x})}{\mathrm{g}_{\mathrm{m}}(\mathbf{y})}\right) \\
&=\mathrm{d}_{a}^{2}(\mathbf{x}, \mathbf{y})+D \log ^{2}\left(\frac{\mathrm{g}_{\mathrm{m}}(\mathbf{x})}{\mathrm{g}_{\mathrm{m}}(\mathbf{y})}\right) \geq \mathrm{d}_{a}^{2}(\mathbf{x}, \mathbf{y})
\end{aligned}
$$

(we wrote an answer in LaTeX but now see that we should provide it in HTML, I hope the formula is ok in this form for you. Otherwise, it can be found in https://doi.org/10.1002/9781119976462.ch13 (page 189, Equation 14.1).

Line 155: a bracket is missing (method bdls_pls [41])

Answer: Thanks, done.

Figure 1 and 3: the name of the elements in the plots are hardly visible; please zoom it, if it is possible, or use different colours

Answer: The size of the scores are now  enlarged for figure 3 and this figure improved. For figure 1: We already used quite divers colours (and different symbols, and alpha-transparency shading). In addition, when we would make the symbols larger in Figure 1, the plot becomes unreadable. Since the plots are produced in vector-based graphics, the users can easily zoom to see the details without loosing quality. 

Discussion

Line 205: I would report only the references “[42]”, instead of “see e.g. [42]”

Answer: Thanks, done.

Line 212: “Especially, for instrumental signals (e.g., NMR, LC-MS, or GC-MS) [44] and especially [45] proposed to use all possible log-ratios before (possible) feature selection of the possible large number of log-ratios, because a centered log-ratio transformation may average too much leading to a higher false discovery rates of biomarkers [45]” this sentence is not very clear as well as the positioning of the references. Please rephrase, if it is possible

Answer: Is rephrased to: In addition, for instrumental signals (e.g., NMR, LC-MS, or GC-MS) also all possible log-ratios may be used  instead of centered or isometric log-ratios (filzmoserWal14). That is, each variable is divided by one of the other variables before the logarithm is taken. For a data set with 10 variables, there are already 45 possible log ratios between the variables. (maly19) suggested using all possible log-ratios. Since this leads to a large number of log-ratios, they suggest feature selection to reduce the number of log-ratios obtained. They argure that a centered log-ratio transformation may \textit{average too much} leading to a higher false discovery rates of biomarkers maly19.

Materials and methods

Section 4.1: I would add more details about the analytical method applied

Answer: Thanks, done.

Line 237: “…analysed by [37]” I would report also the name of the authors, e.g. “…analysed by Liu et al. [37]”

Answer: Thanks, done.

Line 248: “…were selected by [37]” I would report also the name of the authors

Line 249: “…the authors of [37]” I would report also the name of the authors

Line 253: “…data published by [37]” I would report also the name of the authors

Line 255: “…by the authors of [29]” I would report also the name of the authors

Line 262: “…were analysed by [29]” I would report also the name of the authors

Line 264: “…by [29]” I would report also the name of the authors

Line 265: “…data from [29]” I would report also the name of the authors

Answer: Thanks, done, except for 249 where we found it self-explaining.

Line 265: “…was reanalysed that was available as [47]” please clarify

Answer: Thanks. Changed to: and was made available on Mendeley Data \cite{russel19b}

Line 297: “…mg/kg, and “cancel the comma

Answer: This is as correct as your suggestion, see e.g. Grammarly blog. Thus we let it as it is, but if the copyeditors also feel that this is wrong, we are happy to change it.

Line 303 “[8] recognized” I would report also the name of the authors

Answer: Thanks, done.

Line 308: “…the results, and “cancel the comma

Answer: see previous statement on the comma

From Line 305 to 312: too long sentence, please consider to rephrase

Answer: Thanks, done.

Line 346 “…[41], and “cancel the comma

Answer: see previous statement on the comma

Line 347 “…i.e., values below…“cancel the comma

Answer: see previous statement regarding the use of the comma before and

Line 350 “… [41, 55-57], considering “a comma missing after bracket

Answer: Thanks, done.

Line 361: “… [55] found” I would report also the name of the authors

Answer: Thanks, done.

Line 387: “…and (4) on centered log-ratio” instead of “and on (4) centered log-ratio”

Answer: Thanks, done.

Reviewer 3 Report

This manuscript discusses some issues regarding the statistical analysis of elemental composition of food samples, with focus on the authentication of honey and discrimination of saffron samples according to origin. I suggest the authors to revise the text along the following lines.

1) I do not agree with the title. Chemical composition refers to elements and also to molecules (organic compounds present in the food samples), however the paper refers only to elemental composition. This would also probably affect the remainder of the text, wherever appropriate.

2) Section 5.1. Standardization before PCA is well-known. It should be noticed somewhere, with references.

3) Some confusion may arise from the use of the word “classification”. I suggest the authors to consider the following scheme: all qualitative methods can be named as “pattern recognition models”, divided in: (a) exploratory analysis, (b) class-modelling and (c) discrimination.

             Exploratory analysis (PCA, HCA) is used to identify trends, similarities and differences between samples, by reducing the dimensionality of the data and/or extracting dominant patterns in complex matrices.

Class-modelling is when you have a well-defined class, train a model, produce a prediction rule, and proceed to include future samples in this specific class, or to a universe of less well-defined classes (SIMCA, see Wold, S., 1976. Pattern recognition by means of disjoint principal components models. Pattern Recognit. 8, 127–139 and also TrAC Trends in Analytical Chemistry Volume 78, April 2016, Pages 17-22, Discriminant analysis is an inappropriate method of authentication, O. Ye. Rodionova,  A. V.Titov, A. L. Pomerantsev).

Discrimination is when you have two or more well-defined classes, train a model, produce a discrimination rule, and proceed to assign future samples to any of these previously defined classes (PLS-DA, see Barker, M., Rayens, W., 2003. Partial least squares for discrimination. J. Chemom. 17, 166–173).

These three activities should not be confused or mixed up. See, for instance, Qualitative pattern recognition in chemistry: theoretical background and practical guidelines, P. Oliveri, C. Malegori, E. Mustorgi, M. Casale, https://doi.org/10.1016/j.microc.2020.105725: "Inappropriate uses of exploratory methods for predictive purposes. It must be remarked clearly that exploratory methods are not appropriate for making a classification, i.e., for predicting the membership of a new sample to a given class."

Rodionova, O.Y., Oliveri, P., Pomerantsev, A.L., Rigorous and compliant approaches to one-class classification, Chemometrics and Intelligent Laboratory Systems, 2016, 159, pp. 89-96, https://doi.org/10.1016/j.chemolab.2016.10.002

             Rodionova, O.Y., Titova, A.V., Pomerantsev, A.L., Discriminant analysis is an inappropriate method of authentication, TrAC - Trends in Analytical Chemistry, 2016, 78, pp. 17-22, https://doi.org/10.1021/acs.analchem.9b04611

What could happen if a typical class-modelling problem is forced to fit into a discriminant strategy? It is forced into a multi-class problem (the compliant classes are well-defined, while the remaining one includes all non-compliant samples). There are limitations for this approach, considering that the position of the delimiter will be heavily influenced by the distribution of the training samples of the non-target class. The latter one, besides being hardly definable as a class, is often represented in the training set by a statistical sample which is very far from being representative of the corresponding statistical population. This may lead to a biased delimiter and thus to biased predictions on new samples.

In sum, the saffron example is clearly a discrimination problem, and it is OK to use LDA, kNN or ANN. The honey example, on the other hand, is a mixture of discrimination (honey-syrup) and class-modelling (adulterated-non adulterated), although I would prefer to treat it as a class-modelling problem (genuine honey vs. all other non-genuine honey samples). I guess the best way to tackle the honey example is using SIMCA.

4) The English should be revised. I quote below some examples, but there might be other ones throughout the text.

- Abstract: … much analysis …

- Abstract: exemplary means “worthy of imitation; commendable: exemplary behavior”, not the meaning intended here. Perhaps use “As examples, …”

- Abstract: Our study demonstrates that how the PCA … Our study demonstrates how PCA …

- Abstract: Furthermore, (comma needed)

- Abstract: For instance, (likewise)

- Introduction, around line 27: “cluster analysis, principal component analysis (PCA), classification methods, and regression methods [2–4] and partial least squares regression [5,6] are commonly used.” PLS is a regression method. Classification methods includes almost everything except (PCA), see above. And “are commonly used” should be deleted.

- Line 30: it is better to say “Inspection of the relevant literature about …”

- Line 41: potentially wrong, not potential wrong

- Line 61: challange … challenge

- Results, Line 78: explain in the text to what example the figure refers.

As I said, there are many other examples which the authors should polish before publication.

Author Response

This manuscript discusses some issues regarding the statistical analysis of elemental composition of food samples, with focus on the authentication of honey and discrimination of saffron samples according to origin. I suggest the authors to revise the text along the following lines.

Answer: Thank you for the detailed comments that lead to an improvement of the manuscript.

1) I do not agree with the title. Chemical composition refers to elements and also to molecules (organic compounds present in the food samples), however the paper refers only to elemental composition. This would also probably affect the remainder of the text, wherever appropriate.

Answer: But organic compounds might be also compositions. Isn't it just another composition in a data set (amount of organic, non-organic, ...)? However, since our applications were only to element concentrations, we changed the title accordingly.

2) Section 5.1. Standardization before PCA is well-known. It should be noticed somewhere, with references.

Answer: We noticed it and added a reference.

3) Some confusion may arise from the use of the word “classification”. I suggest the authors to consider the following scheme: all qualitative methods can be named as “pattern recognition models”, divided in: (a) exploratory analysis, (b) class-modelling and (c) discrimination.

Answer: I would not use this wording. A discrimination or class-modelling method that is based on artificial neural networks would not be a pattern recognizion model. I think many other examples can be named.

Exploratory analysis (PCA, HCA) is used to identify trends, similarities and differences between samples, by reducing the dimensionality of the data and/or extracting dominant patterns in complex matrices.

Answer: I think it is much more, also visual tools should be named. But I aggree that PCA is an exploratory tool, and we used it as explanatory tool.

Class-modelling is when you have a well-defined class, train a model, produce a prediction rule, and proceed to include future samples in this specific class, or to a universe of less well-defined classes (SIMCA, see Wold, S., 1976. Pattern recognition by means of disjoint principal components models. Pattern Recognit. 8, 127–139 and also TrAC Trends in Analytical Chemistry Volume 78, April 2016, Pages 17-22, Discriminant analysis is an inappropriate method of authentication, O. Ye. Rodionova,  A. V.Titov, A. L. Pomerantsev).

Answer: Still a classification problem/method.

Discrimination is when you have two or more well-defined classes, train a model, produce a discrimination rule, and proceed to assign future samples to any of these previously defined classes (PLS-DA, see Barker, M., Rayens, W., 2003. Partial least squares for discrimination. J. Chemom. 17, 166–173).

Answer: Agree, but ANN is then what? You do not have a discrimination rule for knn either. Many methods can be used for classification, simca or others.

These three activities should not be confused or mixed up. See, for instance, Qualitative pattern recognition in chemistry: theoretical background and practical guidelines, P. Oliveri, C. Malegori, E. Mustorgi, M. Casale, https://doi.org/10.1016/j.microc.2020.105725: "Inappropriate uses of exploratory methods for predictive purposes. It must be remarked clearly that exploratory methods are not appropriate for making a classification, i.e., for predicting the membership of a new sample to a given class."

Rodionova, O.Y., Oliveri, P., Pomerantsev, A.L., Rigorous and compliant approaches to one-class classification, Chemometrics and Intelligent Laboratory Systems, 2016, 159, pp. 89-96, https://doi.org/10.1016/j.chemolab.2016.10.002

Rodionova, O.Y., Titova, A.V., Pomerantsev, A.L., Discriminant analysis is an inappropriate method of authentication, TrAC - Trends in Analytical Chemistry, 2016, 78, pp. 17-22, https://doi.org/10.1021/acs.analchem.9b04611

What could happen if a typical class-modelling problem is forced to fit into a discriminant strategy? It is forced into a multi-class problem (the compliant classes are well-defined, while the remaining one includes all non-compliant samples). 

There are limitations for this approach, considering that the position of the delimiter will be heavily influenced by the distribution of the training samples of the non-target class. The latter one, besides being hardly definable as a class, is often represented in the training set by a statistical sample which is very far from being representative of the corresponding statistical population. This may lead to a biased delimiter and thus to biased predictions on new samples.

Answer: Thank you for these enlightment statements on class modelling.

In sum, the saffron example is clearly a discrimination problem, and it is OK to use LDA, kNN or ANN. The honey example, on the other hand, is a mixture of discrimination (honey-syrup) and class-modelling (adulterated-non adulterated), although I would prefer to treat it as a class-modelling problem (genuine honey vs. all other non-genuine honey samples). I guess the best way to tackle the honey example is using SIMCA.

Answer: We run SIMCA as you suggested for the honey samples. SIMCA gives quite bad results. Also the robust version does not improve the results. From a prediction error point of view it is thus not a method that can compete with other ones like deep artificial neural networks.

The average over all rounded imputation methods, just reported for raw and clr transformed data are for classical and robust simca:

| method | transformation | miss rate  |
|--------|----------------|------------|
| ann    | clr            | 0.08430232 |
| knn    | clr            | 0.16759907 |
| lda    | clr            | 0.19813520 |
| CSimca | clr            | 0.22668998 |
| RSimca | clr            | 0.23531469 |
| ann    | raw            | 0.25872093 |
| knn    | raw            | 0.32622378 |
| lda    | raw            | 0.33053613 |
| CSimca | raw            | 0.45151515 |
| RSimca | raw            | 0.47517483 |

Moreover, the aim of the paper is to show differences between a compositional analysis and a non-compositional one. However, if you are insisting we can easily include this method and include the results (they are ready and already calculated). However, because the results (by far) are not competing, I would just not add them. However, we took a note in the discussion about it and included the reference of Rodionova, O.Y., Oliveri, P., Pomerantsev, A.L. (2016).

4) The English should be revised. I quote below some examples, but there might be other ones throughout the text.

We improved the manuscript but since we are not native speakers we have chosen the service from MDPI. With our submission we clicked on the commercial service from MDPI for language editing. Not sure if you receive this submission or a version after language editing. Please you can assume that the language editing service from MDPI that we paid takes care of English.

- Abstract: … much analysis …

- Abstract: exemplary means “worthy of imitation; commendable: exemplary behavior”, not the meaning intended here. Perhaps use “As examples, …”

- Abstract: Our study demonstrates that how the PCA … Our study demonstrates how PCA …

- Abstract: Furthermore, (comma needed)

- Abstract: For instance, (likewise)

- Introduction, around line 27: “cluster analysis, principal component analysis (PCA), classification methods, and regression methods [2–4] and partial least squares regression [5,6] are commonly used.” PLS is a regression method. Classification methods includes almost everything except (PCA), see above. And “are commonly used” should be deleted.

- Line 30: it is better to say “Inspection of the relevant literature about …”

- Line 41: potentially wrong, not potential wrong

- Line 61: challange … challenge

- Results, Line 78: explain in the text to what example the figure refers.

Answer: All the mentioned issues are resolved using/paying the language editing services of MDPI. As mentioned before, note that this service can be selected only after submisssion, so we are not sure if you will see our submitted version or if you receive the edited version of the commercial MDPI language editing service.

Round 2

Reviewer 3 Report

Authors have responded to the queries and issues raised in reviewing the original version. Still some improvement in the English is required, but according to the authors this will be implemented after submission.

Overall, I am satisfied with the authors' answers.

Regarding ANN, as asked by the authrors in their response, there are exploratory unsupervised tools (Kohonen self-organizing maps), tools for one-class problems (support vector machines) and discrimination tools (perceptron feed-forward ANN regression, much as PLS-DA). However, this may not be a topic for the present paper.